# Predictors of life-threatening complications in relatively lower-risk patients hospitalized with COVID-19

Christopher J. Gonzalez[1]*, Cameron J. Hogan[2], Mangala Rajan[1], Martin T. Wells[2], Monika M. Safford[1], Laura C. Pinheiro[1], Arnab K. Ghosh[1], Justin J. Choi[1], Clare A. Burchenal[1], Pooja D. Shah[1], Martin F. Shapiro[1]

1 Department of Medicine, Weill Medical College of Cornell University, New York, New York, United States of America, 2 Department of Statistics and Data Science, Cornell University, Ithaca, New York, United States of America

* cjg7003@med.cornell.edu

**Data Availability Statement:** The data underlying the results presented in the study are available from Weill Cornell Medicine at arch-support@med.

## Abstract

Older individuals with chronic health conditions are at highest risk of adverse clinical outcomes from COVID-19, but there is widespread belief that risk to younger, relatively lower-risk individuals is negligible. We assessed the rate and predictors of life-threatening complications among relatively lower-risk adults hospitalized with COVID-19. Of 3766 adults hospitalized with COVID-19 to three hospitals in New York City from March to May 2020, 963 were relatively lower-risk based on absence of preexisting health conditions. Multivariable logistic regression models examined in-hospital development of life-threatening complications (major medical events, intubation, or death). Covariates included age, sex, race/ethnicity, hypertension, weight, insurance type, and area-level sociodemographic factors (poverty, crowdedness, and limited English proficiency). In individuals ≥55 years old (n = 522), 33.3% experienced a life-threatening complication, 17.4% were intubated, and 22.6% died. Among those <55 years (n = 441), 15.0% experienced a life-threatening complication, 11.1% were intubated, and 5.9% died. In multivariable analyses among those ≥55 years, age (OR 1.03 [95%CI 1.01–1.06]), male sex (OR 1.72 [95%CI 1.14–2.64]), being publicly insured (versus commercial insurance: Medicare, OR 2.02 [95%CI 1.22–3.38], Medicaid, OR 1.87 [95%CI 1.10–3.20]) and living in areas with relatively high limited English proficiency (highest versus lowest quartile: OR 3.50 [95%CI 1.74–7.13]) predicted life-threatening complications. In those <55 years, no sociodemographic factors significantly predicted life-threatening complications. A substantial proportion of relatively lower-risk patients hospitalized with COVID-19 experienced life-threatening complications and more than 1 in 20 died. Public messaging needs to effectively convey that relatively lower-risk individuals are still at risk of serious complications.

## Background

In the early phase of the novel coronavirus (COVID-19) pandemic, reports highlighted its disproportionate impact on older individuals with chronic medical comorbidities [1–3]. Based

cornell.edu. Architecture for Research Computing in Health (ARCH) has created the COVID Institutional Data Repository (IDR).

**Funding:** CJG was supported by the Diversity Center of Excellence (grant D34HP31879 from the Health Resources and Services Administration), the Clinical and Translational Science Center (grant UL1 TR000457 from the National Institutes of Health), New York–Presbyterian Hospital, and Weill Cornell Medicine. The funders had no role in study design, data collection and analysis, decision to publish, or preparation of the manuscript.

**Competing interests:** The authors have declared that no competing interests exist.

on these published data, the Center for Disease Control and Prevention suggests that older individuals and those with certain underlying medical conditions, including cancer, chronic kidney disease, chronic obstructive pulmonary disease, and type 2 diabetes mellitus, are at increased risk for severe illness from COVID-19 [4]. These reports have contributed to a public perception that COVID-19 is a disease that primarily has severe health consequences for older individuals and those with multiple chronic comorbidities [5–8]. These perceptions may in turn affect behavior: young adults (18–29 years old) had the lowest reported use of COVID-19 mitigation behaviors including social distancing and mask wearing [9].

However, growing evidence suggests that COVID-19 leads to life-threatening complications across a broad range of individual ages and risk factors [10–12]. Acute respiratory failure requiring invasive mechanical ventilation, renal dysfunction requiring renal replacement therapy, and death due to COVID-19 have been observed across patients of all ages and with a variety of comorbidities [1, 13]. Nonetheless, the clinical course of relatively lower-risk patients (those without pre-existing serious medical conditions) hospitalized with COVID-19 has not been delineated. While these patients are widely considered to be at lower risk, numerous factors may contribute to variations in their outcomes. For example, COVID-19 outcomes vary by race, ethnicity, and area-level socioeconomic factors, but these variations have generally been attributed to differences in the prevalence of chronic comorbidities, which themselves vary by race and ethnicity [14–16]. It remains unclear whether demographic and socioeconomic factors also are associated with unfavorable outcomes in relatively lower-risk individuals.

In this study, we studied life-threatening events in relatively lower-risk patients admitted to three hospitals in New York City with COVID-19 between March and May 2020, and the association of sociodemographic factors with these outcomes. We theorized that life-threatening complications were common in relatively lower-risk individuals and we hypothesized that advanced age, ethnoracial minority status, lack of health insurance, area-level poverty, residential crowdedness, and limited English proficiency would be associated with higher likelihood of life-threatening events in this population.

## Methods

### Sample

This retrospective cohort study included adults ≥18 years with confirmed COVID-19 who were admitted to three hospitals in New York City during the early surge of the pandemic (March 1 to May 15th, 2020). Data were recorded and available for an 862-bed academic hospital in Manhattan (NewYork-Presbyterian/Weill Cornell Medical Center) and two affiliates: a 180-bed non-teaching community hospital, also located in Manhattan (NewYork-Presbyterian Lower Manhattan Hospital), and a 535-bed teaching hospital in Queens (NewYork-Presbyterian Queens Hospital). The decision to hospitalize patients was based on the clinical judgement of the admitting physician at the time of presentation and unaffected by this retrospective study. COVID-19 diagnosis was confirmed through reverse-transcriptase polymerase chain reaction assays performed on nasopharyngeal swab specimens. Data regarding clinical outcomes were manually abstracted from electronic health records using a quality-controlled protocol and structured abstraction tool [17]. Participants were followed until discharge, transfer out of the hospital, or in-hospital death.

Classification as relatively lower-risk was based on the absence of preexisting comorbid conditions, defined using the Charlson Comorbidity Index (CCI) [18]. The CCI predicts 10-year survival using age and the presence or absence of 16 health conditions: chronic ischemic heart disease, congestive heart failure, peripheral vascular disease, cerebrovascular disease,

chronic pulmonary disease, dementia, connective tissue disease, peptic ulcer disease, liver disease, diabetes mellitus, hemiplegia, moderate to severe chronic kidney disease, solid tumor, leukemia, lymphoma, and AIDS. Prior studies evaluating health outcomes in patients with or without preexisting comorbid conditions have used the CCI [1, 19, 20]. Consistent with the CCI, hypertension was not one of the relevant comorbid conditions in our analysis. In contrast, obesity is not included as a comorbid condition in the CCI. However, we considered patients with severe obesity (body mass index [BMI] $\geq$35 kg/m$^2$) to have a preexisting comorbid condition because individual studies have consistently shown that severe obesity is independently associated with COVID-19 severity and mortality [21–24]. Patients with BMI $\geq$30 but <35 kg/m$^2$, henceforth referred to as low-risk obesity, were included in our cohort because independent associations with COVID-19 severity have not been consistently established in this group [24, 25]. Preexisting comorbid conditions, as defined by the CCI, were manually extracted from electronic medical records and supplemented with ICD-10-CM billing codes from medical records that indicated patient history at the time of admission (S1 Table) [26]. Billing codes for conditions that developed acutely during hospitalization (such as acute myocardial infarction and stroke) were not considered in the identification of preexisting comorbid conditions.

## Measures

Our goal was to assess life-threatening clinical outcomes from COVID-19 in a relatively lower-risk cohort. Life-threatening complications during hospitalization were defined as new myocardial infarction, heart failure or cardiogenic shock, major arrhythmia, disseminated intravascular coagulation, septic shock, positive blood cultures, renal replacement therapy, confirmed venous thromboembolism, intubation, or death. Secondary outcomes were in-hospital mortality, intubation, and length of hospitalization. Length of hospitalization was defined as days from initial hospitalization to discharge or death. Outcome data were collected by manual abstraction from the electronic medical records. Abstraction of life-threatening complications was adjudicated based on physician documentation and/or diagnostic results within the electronic medical records. Major arrhythmias included physician documentation of atrial fibrillation, supraventricular tachycardias, ventricular fibrillation and/or ventricular tachycardia. Positive blood cultures were directly abstracted from microbiology reports. Confirmation of VTE was based on imaging such as a CT-PE or ultrasound.

To assess demographic and socioeconomic predictors of life-threatening complications among patients with COVID-19, we used the Commission on Social Determinants of Health (CSDH) framework proposed by the World Health Organization, which models the underlying processes that contribute to observed health inequities [27]. This model suggests that socioeconomic context and socioeconomic position affect individual biological and non-biological level factors that in turn impact health inequities. Guided by this framework, candidate covariates for each outcome included age, sex, race and ethnicity, history of hypertension, weight category, and health insurance, all extracted from medical records data, and additional area-level socioeconomic factors obtained from publicly available US Census data linked to patient 5-digit zip codes derived from medical records. These candidate area-level covariates were poverty (percentage of households living below the poverty level), crowdedness (percentage with more than one occupant per room), and limited English proficiency (percentage that report speaking English "not well" or "not at all"). Median income and limited educational attainment initially were considered as potential covariates but removed because of collinearity with other area-level covariates. The Agency for Healthcare Research and Quality has utilized these area-level variables to assess area-level socioeconomic status [28, 29].

## Statistical methods

We used descriptive statistics (counts and proportions) to characterize our sample. We present crude unadjusted outcomes for our relatively lower-risk sample, as well as for subsets of that sample (those without hypertension, low-risk obesity, pregnancy or smoking history, and those <35 years old). Outcomes were available and reported for all relatively lower-risk patients; adjusted models included patients for whom all covariates were available. All analyses were stratified a priori into those <55 and ≥55 years of age to separately discern the impact of COVID-19 on younger and older relatively lower-risk patients. Fifty-five years was selected as a threshold for three reasons: first, it equally distributed the number of relatively lower-risk patients; second, it bisected the age of this sample and; third, it bisects the age range (50–59) at which age begins to add risk to the CCI [18].

We used multivariable logistic regression to model in-hospital development of one or more life-threatening complications. We fit Cox proportional hazard models for mortality (measured as time-to-death) and intubation (measured as time-to-intubation) outcomes, and a gamma generalized linear model with a log link function for the length of hospitalization. Independent variables included individual age, sex, ethnoracial identity (Hispanic, non-Hispanic White, non-Hispanic Black, non-Hispanic Asian, and Other/Unknown); history of hypertension; weight category (normal, BMI <25 kg/m$^2$; overweight, BMI ≥25 but <30 kg/m$^2$; and low-risk obesity, BMI ≥30 but <35 kg/m$^2$); health insurance (private/commercial, Medicare and Medicaid; sample size precluded an analysis of those uninsured); and area-level factors reflecting poverty, crowdedness, and limited English proficiency. A significant bivariate association (p<0.10) was used to select variables that went into the multivariable models for each outcome. Length of hospitalization models were also adjusted for hospital transfer and patient death. All Cox proportional hazard models were tested to ensure they met proportionality assumptions, with stratification as warranted.

Statistical analyses were performed in R version 4.0.2. P<0.05 was considered significant in multivariable analyses. This study followed the Strengthening the Reporting of Observational Studies in Epidemiology (STROBE) reporting guideline cross sectional studies [30]. The study was approved by the Institutional Review Board of Weill Cornell Medicine 20–03021681 with a waiver of informed consent.

## Results

### Proportion of patients who were relatively lower-risk

There were 3766 adults hospitalized with COVID-19. Among these, 963 (25.6%) were classified as relatively lower-risk, including 522 (19.4%) of 2686 patients who were ≥55 years old and 441 (40.8%) of 1080 patients who were <55 years old. Among the relatively lower-risk patients, 513 had neither hypertension nor low-risk obesity (13.6% of the overall sample and 53.3% of the relatively low risk group), including 249 ≥55 years old and 264 <55 years old.

Sociodemographic characteristics and presenting symptoms of the relatively lower risk patients are shown in Table 1. Hispanics comprised 44.2% of all relatively lower-risk patients (52.6% of those <55 years and 37.2% of those ≥55 years). By contrast, they comprised 29.9% of patients at higher risk. There was substantial diversity in the sociodemographic characteristics of the sample but few patients were uninsured.

### Prevalence of life-threatening complications in lower-risk patients

The sample of lower-risk patients experienced 508 life-threatening complications. Among lower-risk patients ≥55 years old, 33.3% had one or more life-threatening complications,

**Table 1. Sociodemographic and clinical characteristics for relatively lower-risk patients hospitalized with COVID-19.**

| Characteristics | | <55 years old | | ≥55 years old | |
|---|---|---|---|---|---|
| Total patients, n | | 441 | | 522 | |
| Age, mean years (SE) | | 43.0 | (9.0) | 67.0 | (10.7) |
| Male, | n (%) | 300 | (68.0%) | 316 | (60.5%) |
| Race and Ethnicity, n (%) | | | | | |
| | Hispanic | 232 | (52.6%) | 194 | (37.2%) |
| | White | 52 | (11.8%) | 114 | (21.8%) |
| | Black | 23 | (5.2%) | 32 | (6.1%) |
| | Asian | 72 | (16.3%) | 116 | (22.2%) |
| | Other | 62 | (14.1%) | 66 | (12.6%) |
| Healthcare worker, n (%) | | 21 | (4.8%) | 18 | (3.4%) |
| Current smoker/vaper, n (%) | | 24 | (5.4%) | 9 | (1.7%) |
| Hypertension, n (%) | | 40 | (9.1%) | 215 | (41.2%) |
| Pregnancy, n (%) | | 15 | (3.4%) | 0 | (0.0%) |
| Weight status, n (%) | | | | | |
| | Overweight [BMI 25–29.9kg/m$^2$] | 196 | (44.4%) | 223 | (42.7%) |
| | Obese [BMI 30–34.9kg/m$^2$] | 152 | (34.5%) | 85 | (16.3%) |
| Health insurance, n (%) | | | | | |
| | Commercial | 145 | (32.9%) | 221 | (42.3%) |
| | Medicaid | 229 | (51.9%) | 106 | (20.3%) |
| | Medicare | 43 | (9.8%) | 179 | (34.3%) |
| | Uninsured | 18 | (4.1%) | 5 | (1.0%) |
| Area-level socioeconomic characteristics, mean (SE) | | | | | |
| | Percent living in poverty | 0.19 | (0.09) | 0.19 | (0.10) |
| | Percent of crowded households | 0.15 | (0.09) | 0.13 | (0.08) |
| | Percent with limited English proficiency | 0.32 | (0.20) | 0.29 | (0.20) |
| COVID Symptoms prior to presentation, n (%) | | | | | |
| | Fever | 354 | (80.3%) | 366 | (70.1%) |
| | Dyspnea | 313 | (71.0%) | 370 | (70.9%) |
| | Cough | 357 | (81.0%) | 407 | (78.0%) |
| | Nausea, vomiting or diarrhea | 174 | (39.5%) | 191 | (36.6%) |
| | Myalgias | 143 | (32.4%) | 144 | (27.6%) |

SE = Standard error, BMI = Body Mass Index

17.4% were intubated, and 22.6% died. Among those <55 years old, 15.0% had at least one life-threatening complication, 11.1% were intubated, and 5.9% died. Mean length of hospital stay was 8.3 days in those <55 years and 10.2 days in those ≥55 years (Table 2).

Among the 513 patients without either hypertension or low-risk obesity, the pattern of serious complications was similar to that of all relatively lower-risk patients. The proportion with any life-threatening complications, death, and intubation were 35.3%, 24.5%, 20.1%, respectively, for those ≥55 years old, and 11.7%, 4.2%, and 8.0%, respectively, for those <55 years old (S2 Table). For this subset of patients, mean hospital length was 10.2 days for those ≥55 years and 8.0 days for those <55 years.

Among the 104 relatively lower-risk patients <35 years old, 8.7% had any life-threatening complication, 5.7% were intubated, and 1.9% died (S2 Table). Mean length of hospitalization in this subset was 6.7 days.

**Table 2. Clinical outcomes for relatively lower-risk patients hospitalized with COVID-19.**

| | | <55 years old | | ≥55 years old | |
|---|---|---|---|---|---|
| Total patients, n | | 441 | | 552 | |
| Any new life-threatening complication†, n (%) | | 66 | (15.0%) | 174 | (33.3%) |
| | Septic shock | 17 | (3.9%) | 37 | (7.1%) |
| | Positive blood culture | 14 | (3.2%) | 20 | (3.8%) |
| | Renal replacement therapy | 13 | (2.9%) | 21 | (4.0%) |
| | Major arrhythmias | 10 | (2.3%) | 29 | (5.6%) |
| | New myocardial infarction | 1 | (0.2%) | 7 | (1.3%) |
| | Heart failure or cardiogenic shock | 7 | (1.6%) | 7 | (1.3%) |
| | Confirmed venous thrombus embolism | 9 | (2.0%) | 24 | (4.6%) |
| | Disseminated intravascular coagulation | 3 | (0.7%) | 5 | (1.0%) |
| Intubated, n (%) | | 49 | (11.1%) | 91 | (17.4%) |
| | Time-to-intubation in days, mean (SE) | 4.6 | (5.1) | 4.8 | (4.2) |
| Death, n (%) | | 26 | (5.9%) | 118 | (22.6%) |
| | Time-to-death in days, mean (SE) | 18.0 | (15.1) | 11.7 | (9.4) |
| Length of hospitalization in days, mean (SE) | | 8.3 | (10.5) | 10.2 | (10.7) |

SE = Standard error

†New life-threatening complications include intubation or death

## Multivariable results for life-threatening complications

Results from multivariable analyses are shown in Tables 3 and 4. Among those ≥55 years, age (OR 1.03 [95%CI 1.01–1.06], p<0.01), male sex (OR 1.72 [95%CI 1.14–2.64], p = 0.01), being publicly insured (compared to commercial insurance: Medicare, OR 2.02 [95%CI 1.22–3.38], p = 0.007; Medicaid, OR 1.87 [95%CI 1.10–3.20], p = 0.02), and living in areas with the highest levels of limited English proficiency (highest quartile compared to lowest: OR 3.50 [95%CI 1.74–7.13], p<0.001) were associated with life-threatening complications. In contrast, only the presence of hypertension was significantly associated the occurrence of any life-threatening complication in those <55 years (OR 2.41 [95%CI 1.07–5.21], p = 0.03).

## Multivariable results for intubation and mortality

Age (HR 1.05 [95%CI 1.02–1.08], p = 0.001) and area-level limited English proficiency (HR 3.08 [95%CI 1.51–6.32], p<0.02) was associated with mortality in those ≥55 years. There were no significant individual or area-level associations with mortality in those <55 years.

In those ≥55 years, Hispanic ethnicity was predictive of intubation (HR 2.68 [95%CI 1.32–5.41], p = 0.006). However, there were no significant individual or area-level associations with intubation in those <55 years.

## Multivariable results for length of hospitalization

Among those ≥55 years, no individual or area-level predictors were significantly associated with length of hospitalization. Among those <55 years, age (regression coefficient (β) 0.016 [95%CI 0.004–0.027]; p<0.007), ethnicity (compared to non-Hispanic White: Hispanic, β 0.39 [95%CI 0.04–0.71], p<0.03; Asian, β 0.44 [95%CI 0.04–0.82], p = 0.03; Black, β 0.59 [95%CI 0.07–1.14], p<0.03) and weight status (compared to normal weight: overweight, β 0.33 [95%CI 0.05–0.60], p<0.02; low-risk obesity, β 0.38 [95%CI 0.08–0.67], p = 0.01) were associated with longer hospitalizations.

**Table 3. Multivariable analysis of unfavorable outcomes in relatively lower-risk patients ≥55 years old hospitalized with COVID-19†.**

| | | Any new life-threatening complication | | Intubation | | Mortality | | Length of Hospitalization‡ | |
|---|---|---|---|---|---|---|---|---|---|
| | | OR | (95% CI) | HR | (95% CI) | HR | (95% CI) | β | (95% CI) |
| Age | | 1.03* | (1.01–1.06) | 0.98 | (0.95–1.00) | 1.05* | (1.02–1.08) | -0.01 | (-0.02–0.00) |
| Male | | 1.72* | (1.14–2.64) | 1.36 | (0.85–2.19) | 1.29 | (0.84–1.97) | 0.18 | (-0.01–0.37) |
| Race and Ethnicity | | | | | | | | | |
| | White | 1 | | 1 | | 1 | | 1 | |
| | Hispanic | 1.16 | (0.62–2.16) | 2.68* | (1.33–5.41) | 1.18 | (0.67–2.09) | 0.03 | (-0.23–0.28) |
| | Black | 0.55 | (0.17–1.53) | 0.96 | (0.21–4.40) | 0.38 | (0.09–1.63) | -0.21 | (-0.60–0.21) |
| | Asian | 1.03 | (0.53–2.00) | 2.03 | (0.95–4.32) | 0.88 | (0.49–1.60) | 0.15 | (-0.13–0.43) |
| | Other | 1.22 | (0.59–2.47) | 1.99 | (0.82–4.84) | 0.88 | (0.42–1.85) | -0.03 | (-0.35–0.30) |
| Hypertension | | -- | -- | 0.85 | (0.53–1.36) | -- | -- | -- | -- |
| Weight status | | | | | | | | | |
| | Normal Weight (BMI <25 kg/m²) | 1 | | 1 | | 1 | | 1 | |
| | Overweight (BMI 25–29.9 kg/m²) | 0.94 | (0.60–1.47) | 1.04 | (0.65–1.67) | 0.77 | (0.50–1.17) | 0.06 | (-0.14–0.27) |
| | Obese (BMI 30–34.9 kg/m²) | 0.77 | (0.40–1.47) | 0.77 | (0.39–1.51) | 0.59 | (0.30–1.15) | 0.10 | (-0.19–0.40) |
| Health Insurance§ | | | | | | | | | |
| | Private | 1 | 1 | -- | -- | -- | -- | -- | -- |
| | Medicaid | 1.87* | (1.10–3.20) | -- | -- | -- | -- | -- | -- |
| | Medicare | 2.02* | (1.22–3.38) | -- | -- | -- | -- | -- | -- |
| Area-level socioeconomic characteristics‖ | | | | | | | | | |
| | Percent with limited English proficiency | 3.50* | (1.74–7.13) | -- | -- | 3.13* | [1.16–8.47) | -- | -- |

BMI = body mass index

*p<0.05

†A significant univariate association (p<0.1) was used to select variables that went into the multivariable model for each outcome. Logistic regression was used to model any life-threatening complication, Cox proportional hazard models for mortality and intubation outcomes, and a gamma generalized linear model with a log link function for the length of hospitalization.

‡Also adjusted for intra-institutional transfer and in-hospital death

§Sample size precluded analyses of those without insurance

‖Areas within the highest quartile relative to those within the lowest quartile

## Discussion

This study described the clinical outcomes of relatively lower-risk patients admitted with COVID-19 during the initial phase of the pandemic, and assessed demographic and socioeconomic factors associated with unfavorable outcomes in this group. Despite being considered

**Table 4. Multivariable analysis of unfavorable outcomes in relatively lower-risk patients <55 years old hospitalized with COVID-19†.**

| | Any new life-threatening complication | | Intubation | | Mortality | | Length of Hospitalization‡ | |
|---|---|---|---|---|---|---|---|---|
| | OR | (95% CI) | HR | (95% CI) | HR | (95% CI) | β | (95% CI) |
| Age | 1.02 | (0.99–1.06) | 1.03 | (0.99–1.07) | 1.02 | (0.97–1.08) | 0.02* | (0.00–0.03) |
| Male | 1.66 | (0.90, 3.22) | 1.33 | (0.68–2.64) | 1.57 | (0.59–4.22) | 0.13 | (-0.09–0.34) |
| Race and ethnicity | | | | | | | | |
| White | 1 | | 1 | | 1 | | 1 | |
| Hispanic | 2.15 | (0.80, 7.53) | 1.47 | (0.44–4.86) | 0.78 | (0.17–3.53) | 0.39* | (0.04–0.72) |
| Black§ | 1.77 | (0.31, 9.03) | -- | -- | -- | -- | 0.59* | (0.07–1.14) |
| Asian | 2.03 | (0.63, 7.84) | 2.07 | (0.55–7.71) | 0.50 | (0.08–2.93) | 0.44* | (0.04–0.83) |
| Other | 1.81 | (0.54, 7.16) | 1.61 | (0.54–6.30) | 0.16 | (0.01–1.86) | 0.22 | (-0.18–0.61) |
| Hypertension | 2.41* | (1.07–5.21) | 1.57 | (0.72–3.42) | -- | -- | -- | -- |
| Weight status | | | | | | | | |
| Normal Weight (BMI <25 kg/m²) | 1 | | 1 | | 1 | | 1 | |
| Overweight (BMI 25–29.9 kg/m²) | 1.20 | (0.55, 2.78) | 1.70 | (0.62–4.66) | 0.83 | (0.22–3.12) | 0.33* | (0.05–0.60) |
| Obese (BMI 30–34.9 kg/m²) | 1.54 | (0.69, 3.67) | 2.47 | (0.88–6.92) | 1.32 | (0.36–4.93) | 0.38* | (0.08–0.67) |
| Area-level socioeconomic characteristics‖ | | | | | | | | |
| Percent living in poverty | -- | -- | -- | -- | -- | -- | -0.27 | (-0.58–0.04) |

BMI = body mass index

*p<0.05

†A significant bivariate association (p<0.1) was used to select variables that went into the multivariable model for each outcome. Logistic regression was used to model any life-threatening complication, Cox proportional hazard models for mortality and intubation outcomes, and a gamma generalized linear model with a log link function for the length of hospitalization.

‡Adjusted for intra-institutional transfer and in-hospital death

§Sample size precluded time-to-event analyses of Black individuals <55 years old

‖Areas within the highest quartile relative to those within the lowest quartile

relatively lower-risk, nearly one in seven patients with no preexisting health conditions <55 years of age, and nearly one in three of those ≥55 years of age had a life-threatening complication. When individuals with hypertension and BMIs ≥30 kg/m² were excluded, outcomes remained similar: more than one in nine among those <55 years of age and more than one in three of those ≥55 years old experienced life-threatening outcomes. Similarly, life-threatening complications were alarmingly common among the youngest patients: more than one in twelve among relatively lower-risk patients <35 years of age. While overall case-fatality rates have decreased over the course of the pandemic, COVID-19 remains prevalent in the United States, and it remains unclear how mortality has changed among specific age groups and those without comorbidities [31, 32]. Our findings call attention to the potential for very substantial

risk to health across all populations, and the urgent need to recalibrate public perceptions regarding risks of COVID-19 for even relatively lower-risk.

In contrast to recent narratives suggesting that young and relatively healthy individuals are virtually unaffected by COVID-19 [8, 33], our findings suggest that when they do get sick enough to require hospitalization, they face serious risks of poor outcomes. Over a quarter of all patients admitted to our hospital system with COVID-19 were <55 years old, and over 40% of those were relatively lower-risk according to our classification. This reflects prior data from New York City suggesting that many patients admitted with COVID-19 are relatively young: in one cohort, 45% of patients were between the ages of 20 and 60 years [1]. While that study presented the proportions of death and intubation across several age groups, it reported these outcomes independently of whether patients had any medical conditions and only up until April 4, 2020, early in the pandemic when nearly half of the sample was still hospitalized. We restricted the sample analyzed here to individuals considered to be at relatively lower-risk: patients with a BMI <35 kg/m$^2$ and without major comorbidities, hospitalized throughout the first major surge of the pandemic, and followed until discharge or in-hospital death. Moreover, our findings reveal that many in this group experienced life-threatening complications beyond those who died and/or were intubated.

Differing from trends observed in unadjusted analyses, we did not find increasing age to be associated with unfavorable outcomes among those <55 years old [1, 2]. In this younger cohort, only hypertension was associated with the incidence of life-threatening complications, while no sociodemographic characteristics were significantly associated with mortality or intubation. Hypertension was not associated with intubation or death in any age group. Age, ethnoracial identity, and weight status all were associated with length of hospitalization in the younger relatively lower-risk sample, but not with any other outcomes. The more protracted hospitalizations in these groups may indicate a more severe clinical course or greater deconditioning among these groups, although without any major complications, or alternatively, may reflect unmeasured barriers to discharge. Future research should aim to identify factors prolonging hospitalization in these groups. The lack of any significant association between low-risk obesity and outcomes such as mortality or intubation suggests that previously observed associations between obesity and these outcomes may have been driven by those with BMIs ≥35 kg/m$^2$ [17, 34].

Hispanics comprised a larger portion of relatively lower-risk patients than of all patients admitted with COVID-19. Further research is needed to substantiate this observation and understand potential causes, but it suggests that Hispanics may have been more likely to be hospitalized when relatively lower-risk. Hispanics are more likely to live in overcrowded and multigenerational households [35, 36], factors that often preclude the ability to isolate at home, and increase the risk of infecting other family members. Relatively lower-risk Hispanic patients ≥55 years had higher rates of intubation and longer hospitalizations than non-Hispanic whites, reflecting high severity of illness. This is consistent with recent evidence in which Hispanic patients admitted with COVID-19 had lower average oxygen saturations on presentation than non-Hispanic whites, although that sample did not exclude patients with comorbid conditions or obesity [14]. Notably, living in areas with relatively greater limited English proficiency, but not other sociodemographic characteristics, was associated with life-threatening complications and death in those ≥55 years old. Of the 8.5 million people living in New York City, over a third are foreign born, and nearly a quarter have limited English proficiency; areas with a relatively high density of immigrant residents often have a relatively high concentration of limited English proficiency [37]. Together, these associations suggest the possibility that linguistic structural barriers, or delays in seeking care due to immigration policies may have impacted access to care in the context of COVID-19 in New York City.

Our study has notable strengths, including the diverse sample, rigorously abstracted data, and representation of patients from three different hospitals. There are several limitations also worth noting. The results may not be generalizable to non-academic health centers or those in non-metropolitan areas. Zip code-level data may under-estimate socioeconomic factors more proximal to the individual. Uninsured patients were relatively underrepresented in our sample, as were Black and American Indian individuals. While race and ethnicity were used in our models as potential predictors of unfavorable outcomes, they should not in themselves be considered risk factors and are more likely mediators for unmeasured upstream factors such as racism. Due to inherent limitations in the chart abstraction process, including a lack of complete clinical context, clinical interpretations of blood culture results were not conducted, and potential contaminants may have been included in the analysis. Notably, the prevalence of this complication was low in our sample (3.2% and 3.8% in those <55 and ≥55 years respectively). Public health strategies and clinical care have evolved since accrual of our sample, likely altering the characteristics of patients admitted and decreasing the rates of some outcomes over time. While there are data suggesting that case-mortality has declined since the beginning of the pandemic, future research should assess whether relatively lower-risk patients with COVID-19 continue to experience life-threatening complications [32].

This study found that over a quarter of the people hospitalized with COVID-19 were relatively lower-risk, and that life-threatening complications including death and intubation were common among relatively lower-risk patients hospitalized with COVID-19. While these events were less common among those <55 years of age, their risk was not trivial. While many people with COVID-19 infections experience mild illnesses, people without serious health problems still face risks of very serious complications, including death. It is a matter of great urgency to public health that this message be conveyed clearly to the population.

## Supporting information

**S1 Table. List of comorbid conditions and associated ICD-10 diagnoses codes.**
(DOCX)

**S2 Table. Clinical outcomes for subgroups of relatively lower-risk patients admitted with COVID-19.**
(DOCX)

## Author Contributions

**Conceptualization:** Christopher J. Gonzalez, Mangala Rajan, Martin T. Wells, Monika M. Safford, Laura C. Pinheiro, Arnab K. Ghosh, Justin J. Choi, Martin F. Shapiro.

**Data curation:** Christopher J. Gonzalez, Cameron J. Hogan, Mangala Rajan, Martin T. Wells, Laura C. Pinheiro, Justin J. Choi, Clare A. Burchenal, Pooja D. Shah.

**Formal analysis:** Christopher J. Gonzalez, Cameron J. Hogan, Mangala Rajan, Martin T. Wells, Monika M. Safford, Laura C. Pinheiro, Arnab K. Ghosh, Justin J. Choi, Clare A. Burchenal, Pooja D. Shah, Martin F. Shapiro.

**Investigation:** Christopher J. Gonzalez, Mangala Rajan, Martin T. Wells, Monika M. Safford, Arnab K. Ghosh, Justin J. Choi, Martin F. Shapiro.

**Methodology:** Christopher J. Gonzalez, Cameron J. Hogan, Mangala Rajan, Martin T. Wells, Monika M. Safford, Laura C. Pinheiro, Arnab K. Ghosh, Justin J. Choi, Clare A. Burchenal, Martin F. Shapiro.

**Project administration:** Christopher J. Gonzalez, Cameron J. Hogan, Mangala Rajan, Martin F. Shapiro.

**Resources:** Monika M. Safford, Clare A. Burchenal, Pooja D. Shah.

**Supervision:** Mangala Rajan, Martin T. Wells, Monika M. Safford, Martin F. Shapiro.

**Validation:** Christopher J. Gonzalez, Mangala Rajan.

**Writing – original draft:** Christopher J. Gonzalez, Cameron J. Hogan, Mangala Rajan, Martin T. Wells, Laura C. Pinheiro, Martin F. Shapiro.

**Writing – review & editing:** Christopher J. Gonzalez, Cameron J. Hogan, Mangala Rajan, Martin T. Wells, Monika M. Safford, Laura C. Pinheiro, Arnab K. Ghosh, Justin J. Choi, Clare A. Burchenal, Pooja D. Shah, Martin F. Shapiro.

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
