## [Decision Letter · Decision Letter 0]

8 Oct 2021

PONE-D-21-22788Predictors of Life-Threatening Complications in Relatively Lower-Risk Patients Hospitalized with COVID-19PLOS ONE

Dear Dr. Gonzalez,

Thank you for submitting your manuscript to PLOS ONE. After careful consideration, we feel that it has merit but does not fully meet PLOS ONE’s publication criteria as it currently stands. Therefore, we invite you to submit a revised version of the manuscript that addresses the points raised during the review process.

Please submit your revised manuscript within 60 days of receipt of this email. If you will need more time than this to complete your revisions, please reply to this message or contact the journal office at plosone@plos.org. Please include the following items when submitting your revised manuscript:A rebuttal letter that responds to each point raised by the academic editor and reviewer(s). You should upload this letter as a separate file labeled 'Response to Reviewers'.A marked-up copy of your manuscript that highlights changes made to the original version. You should upload this as a separate file labeled 'Revised Manuscript with Track Changes'.An unmarked version of your revised paper without tracked changes. You should upload this as a separate file labeled 'Manuscript'.

We look forward to receiving your revised manuscript.

Kind regards,

Marlene Camacho-Rivera, ScD, MPH

Academic Editor

PLOS ONE

Journal Requirements:

2. Please include in your Methods section (or in Supplementary Information files) the participating hospitals/institutions

"The work was supported by the Diversity Center of Excellence (grant D34HP31879 from the Health Resources and Services Administration), the Clinical and Translational Science Center (grant UL1 TR000457 from the National Institutes of Health), New York–Presbyterian Hospital, and Weill Cornell Medicine"

"CJG was supported by the Diversity Center of Excellence (grant D34HP31879 from the Health Resources and Services Administration), the Clinical and Translational Science Center (grant UL1 TR000457 from the National Institutes of Health), New York–Presbyterian Hospital, and Weill Cornell Medicine. The funders had no role in study design, data collection and analysis, decision to publish, or preparation of the manuscript" 

4. We note that you have included the phrase “data not shown” in your manuscript. Unfortunately, this does not meet our data sharing requirements. PLOS does not permit references to inaccessible data. We require that authors provide all relevant data within the paper, Supporting Information files, or in an acceptable, public repository. Please add a citation to support this phrase or upload the data that corresponds with these findings to a stable repository (such as Figshare or Dryad) and provide and URLs, DOIs, or accession numbers that may be used to access these data. Or, if the data are not a core part of the research being presented in your study, we ask that you remove the phrase that refers to these data

Reviewers' comments:

Reviewer's Responses to Questions

**Comments to the Author**

1. Is the manuscript technically sound, and do the data support the conclusions?

Reviewer #1: Yes

2. Has the statistical analysis been performed appropriately and rigorously? 

Reviewer #1: I Don't Know

3. Have the authors made all data underlying the findings in their manuscript fully available?

Reviewer #1: Yes

4. Is the manuscript presented in an intelligible fashion and written in standard English?

Reviewer #1: Yes

5. Review Comments to the Author

Reviewer #1: 1. General comments

The authors conducted retrospective cohort studies to assess the rate and predict life-threatening complications among relatively lower-risk adults hospitalized with COVID-19. Though this study is limited, I agree that this cohort, which focused on the low-risk population, is beneficial. However, I have some concerns that should be addressed.

2. Specific comments

① As shown below, I think you need to exclude from the study all those that have been proven to be a risk in at least one meta-analysis.

・Pregnancy and recent pregnancy

・Smoking, current and former

https://www.cdc.gov/coronavirus/2019-ncov/hcp/clinical-care/underlyingconditions.html

② The CDC webpage indicates that obesity as a risk for severe COVID-19 illness is a BMI ≧ 30 kg/m2. Therefore, in your manuscript, you should describe why you did not adopt this result and defined BMI ≧ 35 kg/m2 as obesity.

③ COVID-19 illness is a disease in which a high probability of intubation even without complication. However, the intubation rate is low compared to life-threatening complications in this study, and it is questionable whether all these complications were really life-threatening complications.

What kind of arrhythmia is included?

Were there any false positives or contamination in the blood cultures?

How did you confirm venous thrombus embolism?

You should show details of how these illnesses were diagnosed in this study.

④ The mortality rate of COVID-19 in this population is higher than the general mortality rate (PMID: 33007452). Therefore, you should explain more details about this cohort, such as indications for hospitalization.

⑤ According to this study, living in areas with limited English proficiency was associated with life-threatening complications. This may be a localized problem, and many readers are unfamiliar with New York City, so they may not share this issue. In order to provide beneficial information, please describe the characteristics of the limited English proficiency in New York City.

6. PLOS authors have the option to publish the peer review history of their article (what does this mean?). If published, this will include your full peer review and any attached files.

Reviewer #1: No

---

## [Author Response · Author response to Decision Letter 0]

30 Nov 2021

Dear Editor, 

Thank you for reviewing our manuscript and for your thoughtful feedback. We have addressed the limitations noted within the review, further clarifying our rationale for the inclusion and exclusion criteria used within the study, and have elaborated on context of the study findings. Please find a response to each point raised in the review.

1. As shown below, I think you need to exclude from the study all those that have been proven to be a risk in at least one meta-analysis. (https://www.cdc.gov/coronavirus/2019-ncov/hcp/clinical-care/underlyingconditions.html)

a. Pregnancy and recent pregnancy

b. Smoking, current and former

RESPONSE: 

With regard to the exclusion criteria for our sample, classification as “relatively lower-risk” was based on the absence of preexisting comorbid conditions, defined using the Charlson Comorbidity Index (CCI), which itself predicts 10-year survival using age and the presence or absence of 16 health conditions, rather than on the CDC list of underlying medical conditions associated with higher risk for severe COVID-19, which has continued to evolve over the course of the pandemic. Smoking and pregnancy are not conditions in the Charlson Index. As a result, relatively-lower risk adults with a history of smoking (n=33) or pregnancy (n=15) were included in our main analytic sample (n=963). The number of pregnant individuals in our sample has been added to Table 1.

In Supplementary Appendix 2 (now S2 Table) of our original manuscript, we reported an additional analysis that expanded our exclusion criteria based on existing literature regarding risk factors for COVID-19 severity and mortality. That analysis excluded patients with hypertension and any severity of obesity (BMI >30mg/k2). Based on reviewer comments, the revised manuscript now also excludes patients with a history of smoking or with pregnancy from the relatively low-risk sample in this additional analysis (S2 Table). We have clarified this change in the statistical methods section of the revised manuscript (Page 6). 

2. The CDC webpage indicates that obesity as a risk for severe COVID-19 illness is a BMI ≧ 30 kg/m2. Therefore, in your manuscript, you should describe why you did not adopt this result and defined BMI ≧ 35 kg/m2 as obesity.

RESPONSE:

While prior meta-analyses cited by the CDC have concluded that obesity is associated with the development of critical condition in COVID-19 patients, these studies have noted that the BMI ranges at which the association are observed have varied considerably between individual studies, and have predominantly been observed in ranges over 35 kg/m2. (PMID 32686331) Notably, none of the four individual studies included in that meta-analysis showed a statistically significant association between obesity and severe COVID-19 when using a BMI range of 30-35 kg/m2 to define obesity. Furthermore, this meta-analysis assessed the relation of obesity to critical condition, rather than the risk of mortality. In the revised manuscript, we have clarified the reasoning for using this selection criteria, stating “we considered patients with severe obesity ([BMI] ≥35 kg/m2) to have a preexisting comorbid condition because individual studies have consistently shown that severe obesity is independently associated with COVID-19 severity and mortality. Patients with BMI ≥30 but <35 kg/m2, referred to as low-risk obesity, were included in our cohort because independent associations with COVID-19 morbidity and mortality have not been consistently established in this group.” (Page 4)

3. COVID-19 illness is a disease in which a high probability of intubation even without complication. However, the intubation rate is low compared to life-threatening complications in this study, and it is questionable whether all these complications were really life-threatening complications. You should show details of how these illnesses were diagnosed in this study.

a. What kind of arrhythmia is included?

b. Were there any false positives or contamination in the blood cultures?

c. How did you confirm venous thrombus embolism?

RESPONSE: 

In regards to outcomes observed in our sample, specifically the rate of life-threatening complications, our primary outcome was a composite score of life-threatening complications that included intubation and death. Consistent with reviewer comments about the high probability of intubation without other complication in COVID-19, intubation and death were in fact the most common complications observed in our sample and contributed to a large proportion of all life-threatening complications observed. In regards to details about how illnesses were diagnosed in this study, outcome data were collected by manual abstraction from the electronic medical records, as noted in our methods. We have provided further details about how these outcomes were abstracted, including the following statement: "Abstraction of life-threatening complications was adjudicated based on physician documentation and/or diagnostic results within the electronic medical records. Major arrhythmias included physician documentation of atrial fibrillation, supraventricular tachycardias, ventricular fibrillation and/or ventricular tachycardia. Positive blood cultures were directly abstracted from microbiology reports. Confirmation of VTE was based on imaging such as a CT-PE or ultrasound." (Page 5) Moreover, we cited an additional limitation in the limitations section of our manuscript: “Due to inherent limitations in the chart abstraction process, including a lack of complete clinical context, clinical interpretations of blood culture results were not conducted, and potential contaminants may have been included in the analysis. Notably, the prevalence of this complication was low in our sample (3.2% and 3.8% in those <55 and ≥55 years respectively).” (Page 14) 

4. The mortality rate of COVID-19 in this population is higher than the general mortality rate (PMID: 33007452). Therefore, you should explain more details about this cohort, such as indications for hospitalization.

RESPONSE: 

The mortality rate of COVID19 has certainly varied across populations, as is reflected in this interesting reference. However, the mortality rate observed in our paper is the mortality rate in a relatively lower-risk patient population hospitalized with COVID-19. It does not represent the fatality rate, which would include the number of deaths among all those in the entire population that may be potentially infected. The mortality rate in our study approaches those seen among other patients hospitalized with COVID-19 during similar time periods.(PMID 32320003) The mortality rate in our sample was lower than other hospitalized cohorts given the relative health of this cohort (they are by definition relatively lower-risk). Regarding the indication for hospitalization in our study, we have added a statement to our manuscript stating that the decision to hospitalize patients was based on the clinical judgement of the admitting physician at the time of presentation and unaffected by this retrospective study.(Page 4) 

5. According to this study, living in areas with limited English proficiency was associated with life-threatening complications. This may be a localized problem, and many readers are unfamiliar with New York City, so they may not share this issue. In order to provide beneficial information, please describe the characteristics of the limited English proficiency in New York City.

RESPONSE: 

We have expanded on the linguistic demographics of New York City within our discussion (Page 14), describing the characteristics of limited English proficiency in New York City. 

1. Reviewer Comment: We note that you have included the phrase “data not shown” in your manuscript. Unfortunately, this does not meet our data sharing requirements. PLOS does not permit references to inaccessible data. We require that authors provide all relevant data within the paper, Supporting Information files, or in an acceptable, public repository. Please add a citation to support this phrase or upload the data that corresponds with these findings to a stable repository (such as Figshare or Dryad) and provide and URLs, DOIs, or accession numbers that may be used to access these data. Or, if the data are not a core part of the research being presented in your study, we ask that you remove the phrase that refers to these data

RESPONSE: 

The data is not a core part of the research being presented and the phrase has been removed. 

Thank you for your consideration of our manuscript.

---

## [Decision Letter · Decision Letter 1]

2 Feb 2022

Predictors of life-threatening complications in relatively lower-risk patients hospitalized with COVID-19

PONE-D-21-22788R1

Dear Dr. Gonzalez,

We’re pleased to inform you that your manuscript has been judged scientifically suitable for publication and will be formally accepted for publication once it meets all outstanding technical requirements.

Kind regards,

Marlene Camacho-Rivera, ScD, MPH

Academic Editor

PLOS ONE

Additional Editor Comments (optional):

Reviewers' comments:

Reviewer's Responses to Questions

**Comments to the Author**

1. If the authors have adequately addressed your comments raised in a previous round of review and you feel that this manuscript is now acceptable for publication, you may indicate that here to bypass the “Comments to the Author” section, enter your conflict of interest statement in the “Confidential to Editor” section, and submit your "Accept" recommendation.

Reviewer #1: All comments have been addressed

2. Is the manuscript technically sound, and do the data support the conclusions?

Reviewer #1: Yes

3. Has the statistical analysis been performed appropriately and rigorously? 

Reviewer #1: I Don't Know

4. Have the authors made all data underlying the findings in their manuscript fully available?

Reviewer #1: Yes

5. Is the manuscript presented in an intelligible fashion and written in standard English?

Reviewer #1: Yes

6. Review Comments to the Author

Reviewer #1: (No Response)

7. PLOS authors have the option to publish the peer review history of their article (what does this mean?). If published, this will include your full peer review and any attached files.

Reviewer #1: No

---

## [Editor Report · Acceptance letter]

7 Feb 2022

PONE-D-21-22788R1 

Predictors of life-threatening complications in relatively lower-risk patients hospitalized with COVID-19 

Dear Dr. Gonzalez:

I'm pleased to inform you that your manuscript has been deemed suitable for publication in PLOS ONE. Congratulations! Your manuscript is now with our production department. 

Kind regards, 

on behalf of

Dr. Marlene Camacho-Rivera 

Academic Editor

PLOS ONE